# Nodal lymphangiogenesis and immunophenotypic variations of sinus endothelium in sentinel and non-sentinel lymph nodes of invasive breast carcinoma

Christina C. Westhoff[1]*, Sabrina K. Müller[1], Paul Jank[1], Matthias Kalder[2], Roland Moll[1]

1 Institute of Pathology, Philipps University of Marburg and University Hospital Giessen and Marburg GmbH, Marburg, Germany, 2 Department of Gynecology and Obstetrics, Breast Center Regio, Philipps University of Marburg and University Hospital Giessen and Marburg GmbH, Marburg, Germany

* westhoff@med.uni-marburg.de

**Data Availability Statement:** All relevant data are within the paper and its Supporting Information files. The data set is available via the institutional

## Abstract

Several studies have demonstrated the de novo formation of lymphatic vessels or the reorganization of lymphatic sinus in tumor-draining lymph nodes, partly preceding the detection of lymphatic metastases. This "lymphovascular niche" is supposed to facilitate the survival of metastatic tumor cells. Few studies on nodal lymphangiogenesis in invasive breast cancer (BC) have been published, not considering tumor-free sentinel lymph nodes (SLN) and tumor types. Specimens of SLN and/ or non-SLN (NSLN) of 95 patients with BC were examined immunohistochemically for expression of the lymphatic endothelial marker D2-40 (podoplanin) on lymphatic vessels and the subcapsular sinus. The number of D2-40-positive lymph vessels in metastases was evaluated with two morphometric methods (Chalkley count and number per HPF). Data was explored with respect to TNM parameters, grading, tumor type, size of metastasis, lymph vessel number and hormone receptor/HER2 status with appropriate statistical tests. Lymphangiogenesis was detected exclusively in and around BC metastases with both methods for lymph vessel quantification being equivalent. Lymph vessel number correlated with the size of metastases, being significantly higher in larger metastases (p < 0.001). There was no significant statistical difference with respect to tumor types. Intranodal lymphangiogenesis could not be verified by D2-40 staining in any of the tumor-free lymph nodes examined. However, D2-40 was frequently detected in sinus endothelial/virgultar cells of the subcapsular sinus, partly with strong uniform positivity. Staining intensity and stained proportion of the subcapsular sinus were markedly heterogeneous, significantly correlating with each other both in SLN and NSLN (p < 0.001). A higher proportion of D2-40 stained subcapsular sinus in SLN was significantly associated with worse overall survival (p = 0.0036) and an independent prognostic parameter in multivariate analysis (p = 0.033, HR 2.87). Further studies are necessary to elucidate the biological and clinical significance of the observed immunophenotypic variations of nodal sinus endothelium.

data repository data_UMR of Philipps University Marburg (https://data.uni-marburg.de/handle/dataumr/165.2?locale-attribute=en_US or https://doi.org/10.17192/fdr/99.2).

**Funding:** CCW is supported by the postdoctoral lecture qualification program of the Anneliese Pohl Foundation, Marburg. The funders had no role in study design, data collection and analysis, decision to publish, or preparation of the manuscript.

**Competing interests:** The authors have declared that no competing interests exist.

## Introduction

Invasive breast cancer (BC) is the most frequent malignant tumor in women, leading both in incidence and mortality rate [1]. Lymph node metastasis occurs in about 31 to 38% of cases in Germany [2]. During the 1990s, the concept of sentinel node biopsy was introduced to BC care and has become the standard of care by 2012 with significant reduction of surgical morbidity for the patients [3].

Several studies have demonstrated the de novo formation of lymphatic vessels or the reorganization of lymphatic sinus in tumor-draining lymph nodes, partly preceding the detection of lymphatic metastases [4–9]. However, these studies are mainly relying on animal experiments. This intranodal lymphangiogenesis can be observed before the detection of lymph node metastasis [4–6, 8, 10]–supposedly facilitating the survival of metastatic tumor cells in this "lymphovascular niche" [11]. Only few studies on nodal lymphangiogenesis in human invasive BC have been published [12–14], but these did not examine negative, tumor-free sentinel lymph nodes (SLN) and did not specify tumor types.

D2-40 is a commercially available antibody directed against podoplanin, a 38 kD transmembranous glycoprotein selectively expressed on lymphatic endothelial cells [15–17].

We previously characterized the morphology and immunohistochemical profile of sinus lining and luminal cells and their intercellular junctions in lymph nodes and suggested the term "sinus endothelial/ virgultar cells" for their luminal sinus meshwork [18].

We aimed to examine intranodal lymphovascular changes, including intranodal lymphangiogenesis and subcapsular sinus endothelium regarding the immunohistochemical marker D2-40in a typical patient cohort of a BC center employing the sentinel node biopsy since 2000, stratified by SLN and NSLN, and its potential prognostic impact with respect to overall survival.

Important secondary aspects were the feasibility with routine immunohistochemical repertoire and differentiation for tumor types including invasive lobular carcinoma considering its different tumor biology [19].

## Materials and methods

The archives of the Institute of Pathology, Marburg, were searched for all cases of invasive breast carcinoma (BC) with primary surgery including sentinel node biopsy between 2000 and 2006 and having approved to research purposes (n = 270). Representative paraffin blocks of SLN and/or non-SLN (NSLN) from n = 95 patients were recruited due to complete exhaustion of lymph nodes for routine purposes (Flowcharts for SLN and NSLN in S1 and S2 Figs, respectively). The study was approved by the Ethics Committee of the Medical Faculty of the Philipps University of Marburg (study 57/18). The corresponding clinical and histopathological data was extracted from medical records and pathological reports.

Tissues were fixed in 10% formalin solution, embedded in paraffin, cut at a thickness of 4 µm and stained with hematoxylin and eosin (H&E) for routine purposes. Immunohistochemistry was performed using standard methods (BOND Polymer Refine Detection, Leica, Wetzlar, Germany, with 3,3'-diaminobenzidine [DAB] as chromogen). Podoplanin was detected by the monoclonal antibody clone D2-40, CD31 was detected by the monoclonal antibody JC70A as general endothelial marker for both lymphatic and blood vessels (both antibodies from Agilent Dako, Waldbronn, Germany). The immunostainings were run on an automated immunostaining apparatus (Leica BOND-MAX, Leica).

D2-40 was used as specific marker of the endothelium of (newly formed) lymphatic vessels. Regarding the lymph node sinus, the immunoreactivity of D2-40 was assessed as to the percentage of stained endothelial cells (sinus endothelial/ virgultar cells) of the subcapsular sinus

as continuous and as categorical variables graded in four categories (negative (0%), < 10% of total sinus stained, 10–50% of total sinus stained, 51–80% of total sinus stained and >80% of total sinus stained). Intensity of staining of the endothelial cells in the subcapsular sinus was evaluated as categorical variable (negative, weak, intermediate, strong).

All available SLN and NSLN were studied with one representative lymph node per SLN and/or NSLN if multiple lymph nodes were present. If metastases were present, one representative SLN and/or NSLN with metastasis was analyzed. If multiple metastases were present, the largest metastasis was chosen and its size quantified in millimeters. The number of D2-40-positive lymph vessels in metastases was evaluated in parallel with two established morphometric methods without knowledge of the corresponding clinicopathological findings: On the one hand, the number of D2-40 positive lymph vessels within the metastasis was counted in five high-power fields (HPF, x400) within the hot spots of D2-40 stained vessels and the mean value of vessel counts per HPF was calculated as lymphatic vessel density (LVD) [20]. On the other hand, Chalkley point-overlap morphometry was performed using a Chalkley point array graticule at five spots in 200x magnification within the hot spots of D2-40 stained vessels as detailed in references [21, 22], and the mean value of vessel counts was calculated as Chalkley Count (CC). Areas of fibrosis or hilar vascular structures were excluded.

For statistical analysis, the null hypothesis was "Intranodal lymphangiogenesis can be observed in tumor-draining, but tumor cell-free regional lymph node tissue." The number of lymphatic vessels in SLN and NSLN metastases was calculated and categorized on the basis of their medians in two categories, high and low. For comparison of the two morphometric quantification methods for lymphatic vessels in nodal metastases, a correlation test was performed using Pearson's or Spearman's correlation coefficient, respectively. Statistical analysis was performed using R within RStudio and the packages compareGroups, survivalAnalysis and survminer [23–27]. Differences in immunoreactivity were explored with regard to TNM stage (tumor size in millimeters, pT- and pN-stage), grading, tumor type, size of metastasis, lymph vessel number and hormone receptor/ HER2 status and were evaluated with t-, $\chi^2$- or Fisher's Exact-tests where appropriate. Overall survival (OS) was defined as time from breast surgery until death irrespective of cause and analyzed with respect to histopathological and clinical parameters by Kaplan-Meier curves and log-rank test [24]. For statistically relevant continuous parameters, we used the standardized tool "Cutoff Finder", an algorithm to determine the best cutoff point for OS [28]. Univariate and multivariate Cox regression analysis was carried out regarding OS as dependent variable and age at surgery, pN stage, pT stage and proportion of stained sinus as independent variables.

## Results

The clinicopathological characteristics of the patients involved in this study are displayed in Table 1.

In lymph node metastases of BC, D2-40 staining revealed the presence of lymphatic vessels in the stroma within the metastatic foci or immediately adjacent to them. Among 95 SLN overall investigated, 32 harbored metastases with a size of > 0.2 mm, half of them being micrometastases (S1 Table). Three additional SLN displayed isolated tumor cells (ITC), equivalent to single tumor cells or tumor cell clusters ≤ 0.2 mm and a pN0 stage. In 40 cases, additional NSLN could be investigated comprising eight metastases of > 0.2 mm, three of them being micro- and five being macrometastases. Tumor types were not associated with type of metastasis as detailed in S1 Table. Established lymph node metastases in 20 SLN and five NSLN were available for further immunohistochemical analysis of newly formed D2-40 positive lymphatic vessels, among them four cases of micro- and 16 cases of macrometastasis for SLN as well as 2

**Table 1. Clinicopathological characteristics of the patients involved in this study (n = 95).**

| Parameter | mean or No. of cases (range or % of n) |
|---|---|
| **age, in years** | 55.17 (25–84) |
| **Histological types** | |
| Invasive carcinoma of no special type (NST) | 75 (78.9%) |
| Invasive lobular carcinoma | 18 (18.9%) |
| Tubular carcinoma | 2 (2.1%) |
| **size of the primary tumor, in cm** | 1.46 (0.4–5.1) |
| **pT stage** | |
| pT1 | 80 (84.2%) |
| pT2 | 14 (14.7%) |
| pT3 | 1 (1.1%) |
| **pN stage** | |
| pN0 | 63 (66.3%) |
| pN1 | 28 (29.4%) |
| pN2 | 2 (2.1%) |
| pN3 | 2 (2.1%) |
| **Grading** | |
| well differentiated (G1) | 5 (5.3%) |
| moderately differentiated (G2) | 85 (89.5%) |
| poorly differentiated (G3) | 5 (5.3%) |
| **Estrogen receptor expression** | |
| Negative | 11 (11.7%) |
| Positive | 83 (88.3%) |
| **Progesterone receptor expression** | |
| Negative | 21 (22.3%) |
| Positive | 73 (77.7%) |
| **Her2/neu status** | |
| Negative | 75 (79.8%) |
| Positive | 19 (20.2%) |
| **Triple negative status** | |
| No | 88 (93.6%) |
| Yes | 6 (6.3%) |

cases of micro- and 3 cases of macrometastasis for NSLN. Newly formed D2-40 positive lymphatic vessels within or in immediate vicinity of the metastasis were detected in most cases analyzed albeit in widely different numbers. They were quantified with two different morphometric methods. Quantification by counting the number of D2-40 positive lymphatic vessels per HPF (lymphatic vessel density (LVD)) and by determining the number of Chalkley point-overlaps (CC) was done in parallel in nodal metastases and the possible equivalence of both methods was evaluated. An example of one case is illustrated in Fig 1 showing five hotspots of lymphatic vessels within the metastatic tumor infiltrate quantified as LVD (A1—E1) and CC (A2—E2), respectively.

Both methods for quantification of lymph vessels were found to be equivalent (Fig 2). Spearman's and Pearson's correlation coefficient indicated a strong positive correlation between LVD and CC for SLN (R = 0.98, p < 0.005) and NSLN (R = 0.93, p = 0,021), respectively.

The number of newly formed lymphatic vessels in nodal metastases, categorized on the basis of their medians in high and low, correlated significantly with the size of metastasis in both SLN and NSLN (p < 0.001; Fig 3).

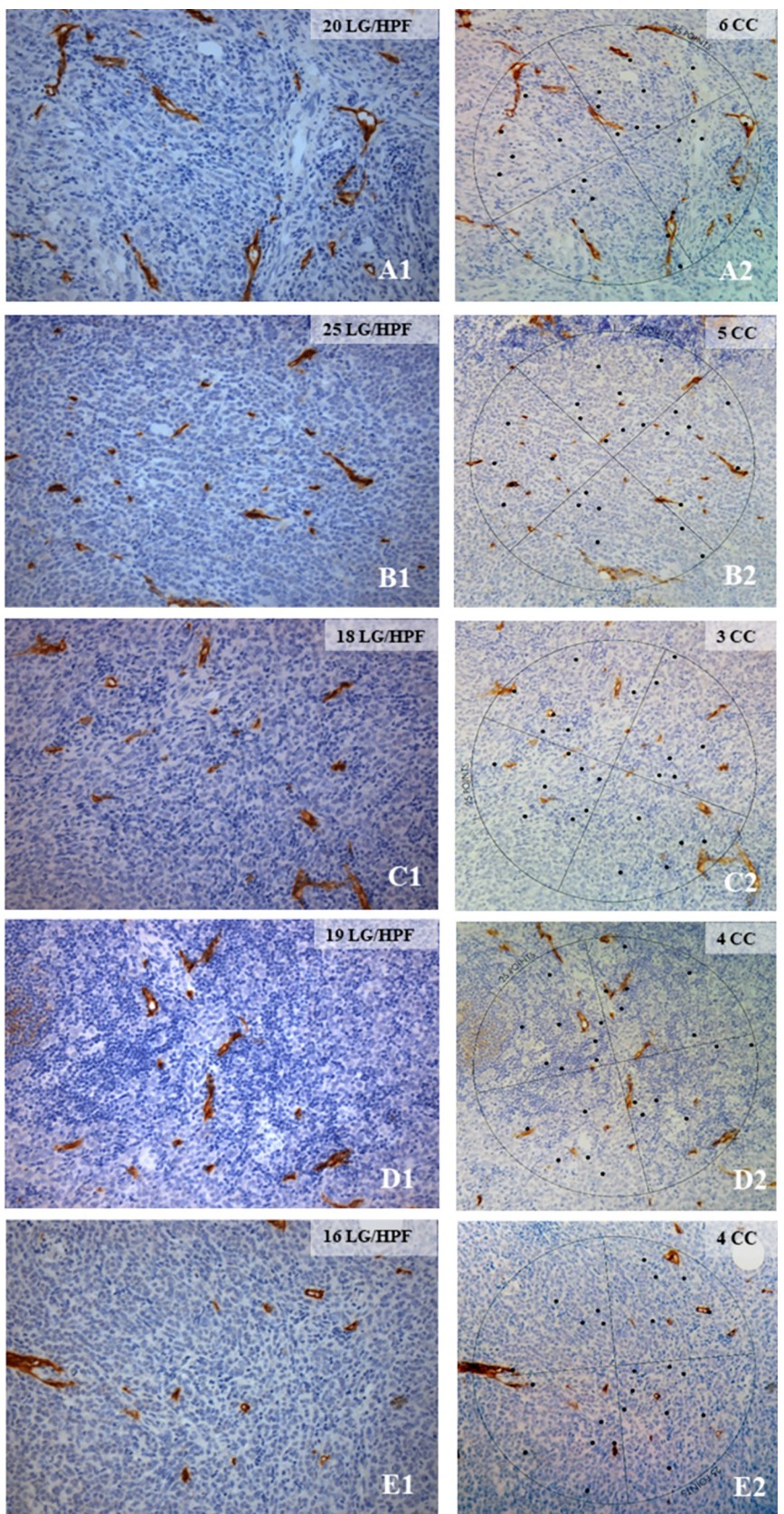

**Fig 1. Illustration of the quantification of newly formed lymph vessels in a SLN metastasis of an invasive breast carcinoma NST with LVD and CC.** The number of D2-40 stained newly formed lymph vessels was counted in five HPF in five different hot spots of D2-40 staining within the metastasis in a SLN, indicated as lymphatic vessels per high power field (LV/HPF) (A1-E1), and the mean was calculated (in Fig 1 19.6 LV/HPF). Correspondingly, the number of points in the Chalkley graticule superposed with D2-40 positive lymph vessels was counted, indicated as CC (A2-E2), and the mean was calculated (in Fig 1 4.4 CC). A1-E1, 400x magnification; A2-E2, 200x magnification.

There was no significant association of the number of lymph vessels in the metastases with respect to other clinical and pathological parameters, including tumor type (S2 Table).

All cases of SLN and NSLN without micro- or macrometastasis in the present study, comprising 63 cases, were meticulously screened for the presence of any typical newly formed lymphatic vessels. However, no such vessels (i.e., small structures with circular cross-sectional profiles) were detected. Thus, intranodal lymphangiogenesis could not be verified in tumor-draining, but tumor cell-free regional lymph node tissue examined in the present study. In addition, D2-40 positive vessels were present in the hilar region of lymph nodes being excluded from analysis because of its locoregional nature.

In the course of our analysis of D2-40 immunostaining in tumor-free lymph nodes, we noticed positive staining present in sinus endothelial/ virgultar cells of preexisting lymph node sinuses (i.e., extended area-like sinus systems containing an intraluminal virgultar reticulum), with predominance of the subcapsular sinus. Such staining was also seen in tumor-free areas of metastatically involved lymph nodes remote from the tumor infiltrate. In the majority of cases, we found specific expression of D2-40 in the sinus endothelial/ virgultar cells of the subcapsular sinus which, however, was remarkably heterogenous. In some lymph nodes, the subcapsular sinus displayed strong D2-40 expression in large parts (Fig 4), whereas in other lymph nodes, the subcapsular sinus showed less staining or was completely negative. Staining intensity and proportion of stained subcapsular sinus for D2-40 correlated in SLN and NSLN ($p < 0.001$), respectively (Fig 5). In contrast, for the general endothelial marker CD31 homogeneous staining of subcapsular sinus endothelium was seen consistently. The extent of subcapsular sinus staining for D2-40 did not correlate with most clinical and histopathological parameters evaluated (S3 Table). However, analysis for OS revealed the proportion of subcapsular sinus in SLN stained for D2-40 as statistically relevant parameter. The standardized "Cut-off Finder" algorithm revealed the best cutoff at 35% with the corresponding Kaplan Meier curve demonstrating better OS for patients with lymph nodes displaying a stained proportion of subcapsular sinus of $\leq$ 35% in SLN ($p = 0.0036$, Fig 6). Multivariate analysis of proportion of D2-40 stained sinus in SLN vs. higher pT stage, positive nodal status and older age at surgery (best cutoff at 63.5 years, determined by standardized"Cutoff Finder" algorithm) revealed that higher proportion of D2-40 positive subcapsular sinus in SLN is an independent prognostic parameter for worse overall survival ($p = 0.033$, Fig 7 and S3 Fig).

## Discussion

Several studies have demonstrated the de novo formation of lymphatic vessels or the reorganization of lymphatic sinus in tumor-draining lymph nodes, partly preceding the detection of lymphatic metastases [4–9]. Hirakawa et al. demonstrated increased numbers of enlarged LYVE-1-sinusoidal vessels in nonmetastasis-containing draining lymph nodes of VEGF-A and VEGF-C overexpressing transgenic mice [4, 8], amended by Liersch et al. for a VEGF-C overexpressing melanoma cell line in nude mice [9]. Harrell et al. also discovered extensive growth of lymphatic sinuses in draining lymph nodes even before tumor cells were detectable in mice injected with melanoma cells [6]. Qian et al. demonstrated reorganization and enlargement of lymphatic sinuses as well as alterations of high endothelial venules (HEV) before

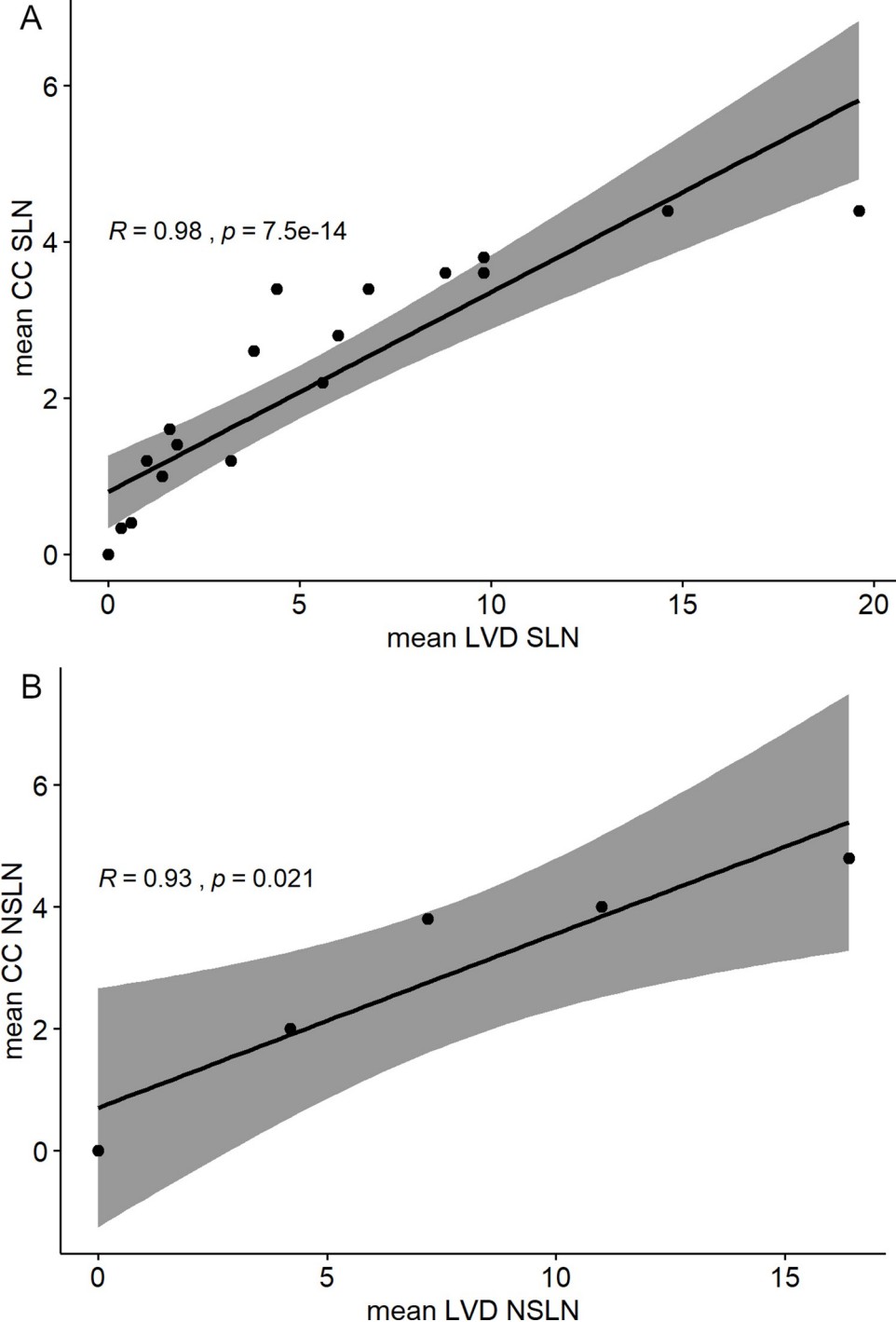

**Fig 2. Correlation of both morphometric quantification methods, lymphatic vessel density and Chalkley count.**
Mean values of lymphatic vessel density (LVD) and Chalkley Count (CC) of all cases analyzed are graphically
illustrated and correlated using Spearman's or Pearson's correlation coefficient, displaying equivalence of both
quantification methods.

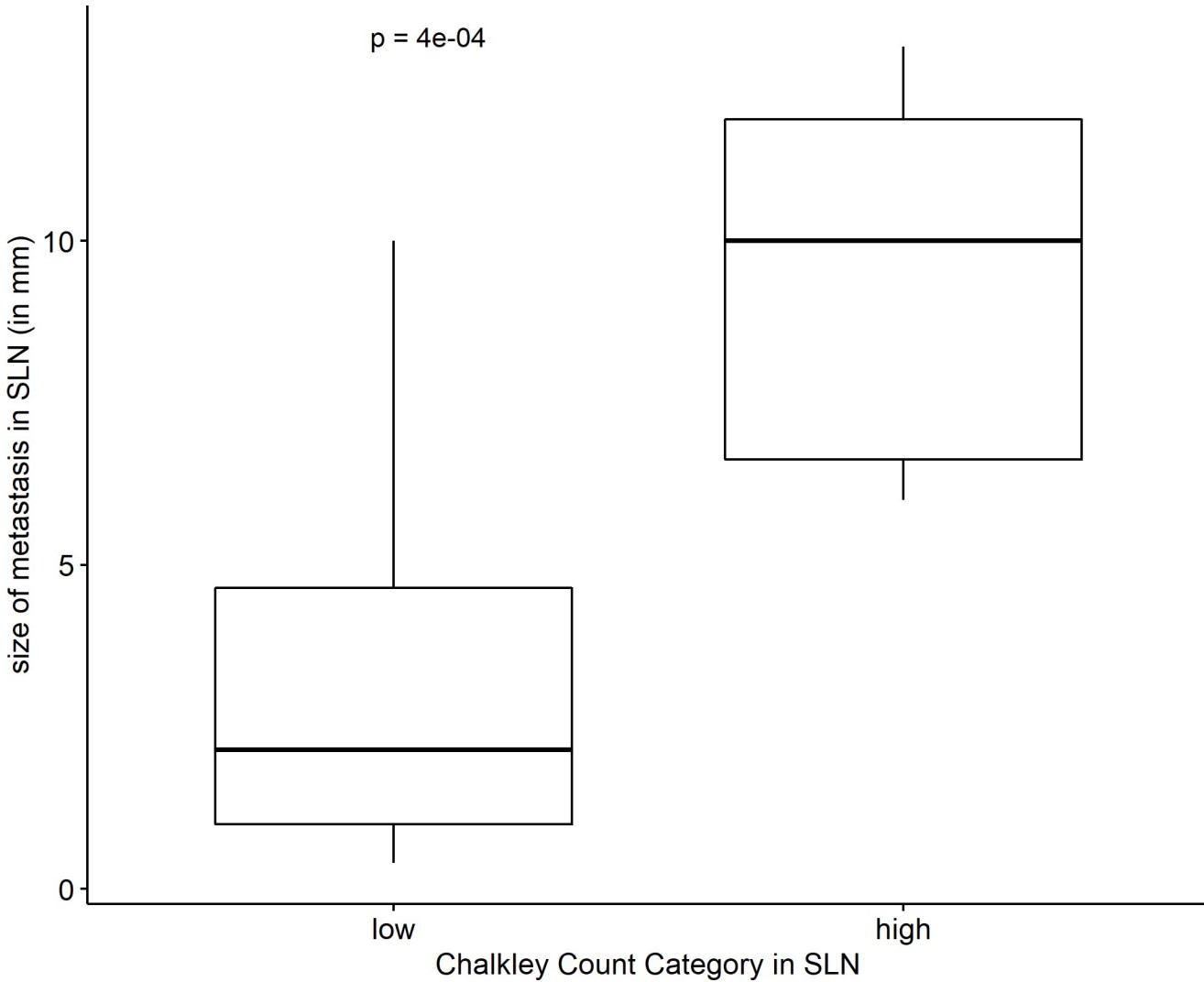

**Fig 3. Correlation of the number of lymphatic vessels with the size of metastasis in SLN.** The number of lymphatic vessels in SLN metastases was calculated and categorized on the basis of the median in two categories, high and low. Here, a box plot displays the Chalkley count category with respect to the size of SLN metastasis in millimeters. The box signifies the first to third quartile with the median as bar. The line indicates all values with the ends representing minimum and maximum values.

metastasis of nasopharyngeal carcinoma and BC cell lines in mice and postulated remodeling of HEV in non-metastatic draining lymph nodes of human BC patients [5].

In the clinical setting, Hirakawa et al. found lymphatic vessel growth with D2-40 in regional lymph nodes of patients with invasive extramammary Paget's disease before effective metastasis, whereas such changes were undetectable in regional lymph nodes of extramammary Paget's disease (in situ), confirmed by quantitative image analysis [10].

Kurahara et al. studied cases of pancreatic adenocarcinoma and found a positive correlation between LVD and nodal metastasis; furthermore, LVD was significantly higher in non-metastatic lymph nodes in patients with nodal metastasis [7].

Jakob et al. found a statistically significant correlation between the presence of intranodal lymphangiogenesis in regional lymph nodes and yUICC stage as well as relapse in rectal cancer patients after neoadjuvant treatment. In addition, intranodal lymphangiogenesis statistically

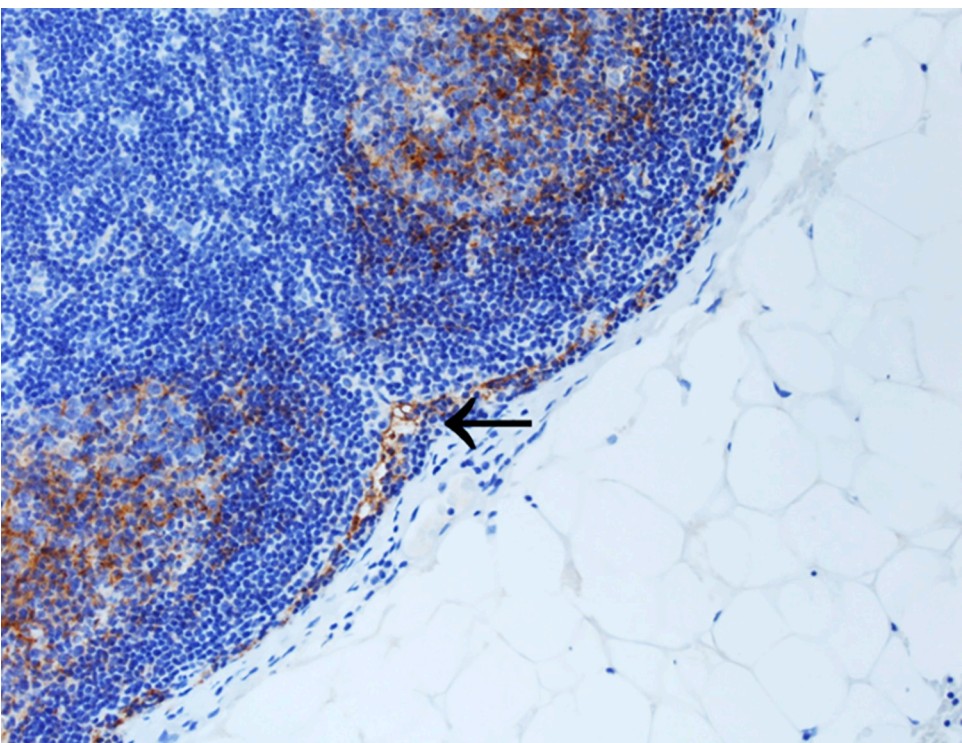

**Fig 4. Staining pattern of D2-40 in the subcapsular sinus of a representative lymph node.** In a SLN without metastasis the subcapsular sinus displays strong staining positivity of the sinus endothelial cells for D2-40 (arrow). Also, positive follicular dendritic cells in the germinal center are depicted (200x magnification).

significantly correlated with disease-free survival (DFS) and was an independent prognostic factor in multivariate analysis. Lymph node lymphangiogenesis also correlated with worse DFS in node-negative patients [20].

For BC, van den Eynden et al. reported lymphatic vessels in most metastatically involved lymph nodes with increased lymphatic endothelial cell proliferation fraction in contrast to uninvolved lymph nodes of BC patients [12]. Increased lymphangiogenesis in SLN metastasis was associated with and an independent predictor of increased frequency of involved NSLN in multivariate analysis [13]. Zhao et al. demonstrated increased Prox-1 and LYVE-1 mRNA in uninvolved SLN of BC patients in comparison to control lymph nodes via quantitative real-time RT-PCR, with higher expression levels in patients with high VEGF-C expressing tumors [14].

With respect to these promising results, we evaluated these findings in a typical patient cohort of a BC center employing the sentinel node biopsy starting in 2000. An important objective was the feasibility with routine immunohistochemical repertoire, as most of the aforementioned studies used double immunofluorescence or double immunohistochemical staining methods not being readily available for routine purposes. We chose podoplanin/ D2-40 as marker since it is well established in routine staining and has been identified as the best marker to visualize lymphatic vessels [29].

Consistent with findings in previous studies in both BC and extramammary cancer [7, 10, 12], D2-40 staining in lymph node metastases of BC revealed the presence of newly formed lymphatic vessels in the stroma within the metastatic foci or immediately adjacent to them, indicating intra- and peritumoral lymphangiogenesis. For rectal cancer patients treated with neoadjuvant radiochemotherapy, Jakob et al. assessed lymph vessel density (LVD) by counting

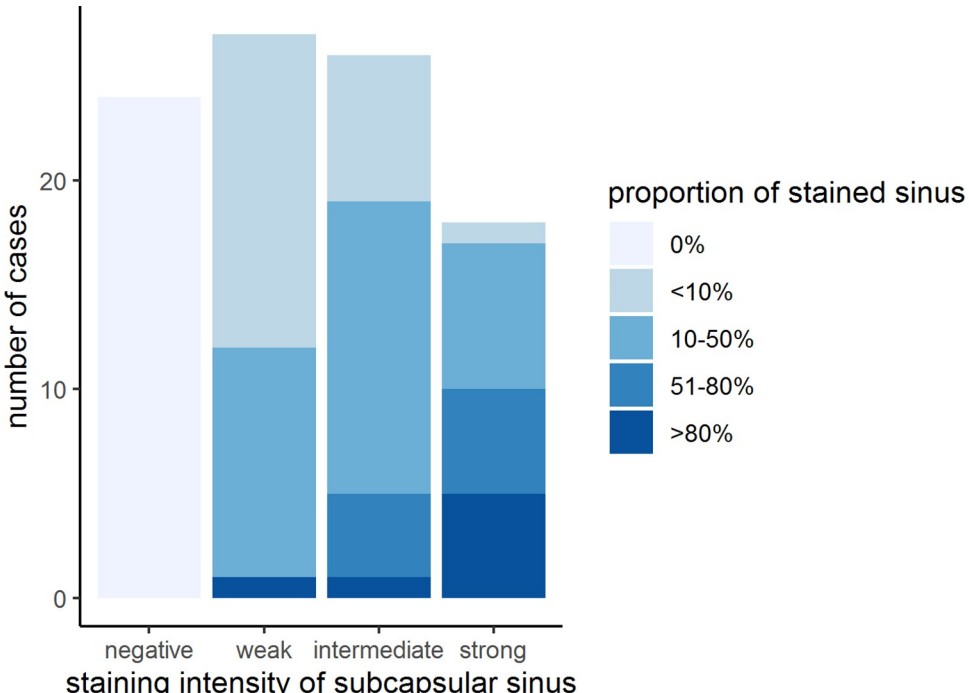

**Fig 5. Correlation of staining intensity and proportion of stained sinus in SLN.** A stacked bar plot visualizes the correlation of staining intensity and proportion of stained subcapsular sinus for D2-40, here within SLN.

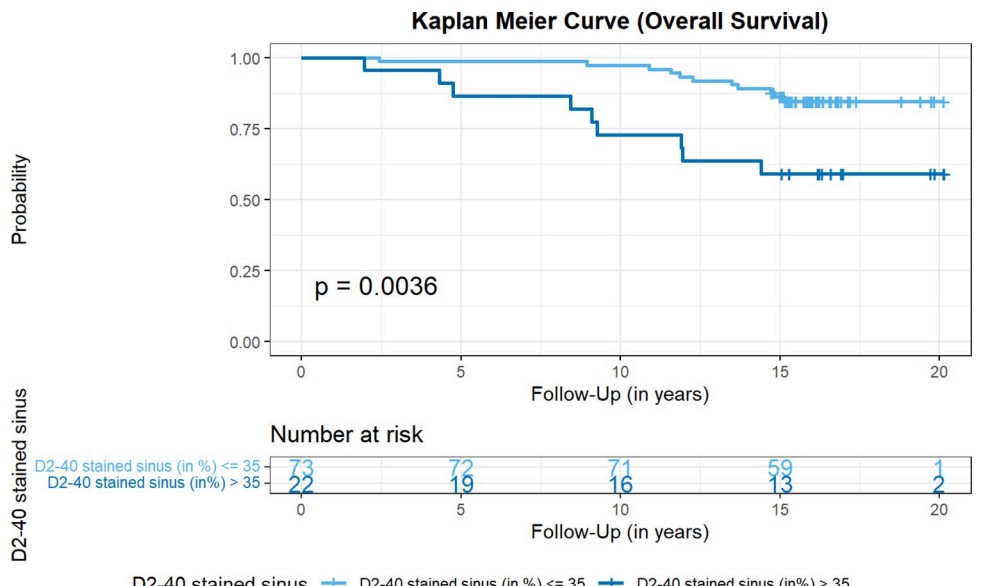

**Fig 6. Kaplan-Meier curve for overall survival with respect to the proportion of D2-40-positive stained sinus in SLN.** According to the optimal cutoff point determined by the Cutoff Finder algorithm, overall survival was plotted for the proportion of D2-40 positive stained subcapsular sinus in sentinel lymph nodes. Overall survival was significantly longer for patients with ≤ 35% stained sinus, p value of the logrank test.

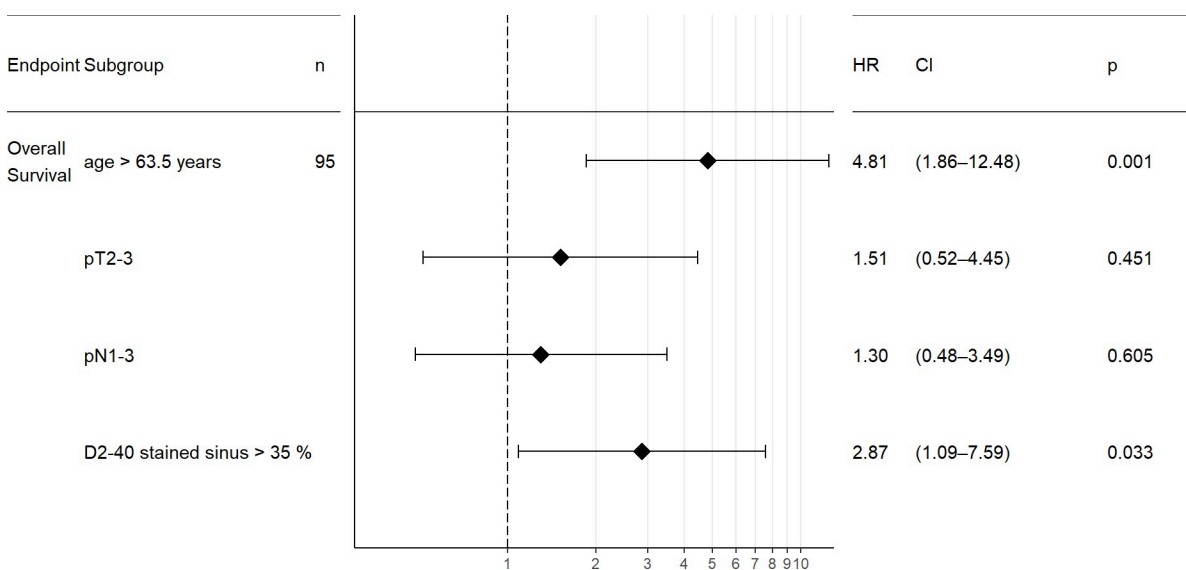

**Fig 7. Forest plot for the multivariate analysis for older age at surgery, higher pT stage, positive nodal status and proportion of D2-40 stained sinus in SLN with respect to overall survival.** The Cox regression hazard ratio (HR), 95% confidence interval (CI) and p values are plotted with respect to overall survival for older patients ("age > 63.5 years") vs. younger patients, higher pT stage ("pT2-3") vs. pT1), positive nodal status ("pN1-3") vs. negative nodal status and higher proportion of D2-40 stained sinus in SLN ("D2-40 stained sinus > 35%") vs. lower proportion of D2-40 stained sinus in SLN. Higher proportion of D2-40 stained sinus in SLN is an independent prognostic parameter for worse overall survival in multivariate analysis.

the number of lymphatic vessels per HPF (400x) as frequently done in routine pathology but also referred to the Chalkley point graticule method [20, 30]. In established lymph node metastases of BC, we here quantified D2-40-positive lymphatic vessels by assessing LVD as well as Chalkley point overlap-morphometry in comparison [21, 22, 30]. Both quantification methods proved to be equivalent. Furthermore, both methods resulted in a statistically significant correlation of LVD and Chalkley count (CC) with the size of nodal metastasis, respectively, with newly formed lymphatic vessels being significantly more numerous in larger metastases, confirming and extending previous studies [7, 10, 12]. Due to the equivalence of both morphometric methods, future studies could be evaluated out only by LVD analysis which is familiar to all clinical pathologists and can be carried out without specialized equipment.

In contrast to the aforementioned literature [4–6, 8, 10, 20], we did not observe any newly formed lymphatic vessels outside established nodal metastases. Most of these studies were based on animal experiments which in part might explain the discrepant findings. Tumors in the mouse models used are frequently overexpressing VEGF-A or -C, promoting lymphangiogenesis and possibly overestimating the effect in comparison to clinical conditions in humans. Also, the time-dependent sequence of nodal metastasis can be better controlled in animal experiments. Thus, in human routine tumor material the time of surgery could present a relevant influence on the detection of newly formed lymphatic vessels and prevent the observation of significant associations. The above-mentioned studies employed a multitude of different lymphatic markers and methodologies for studying lymphangiogenesis, possibly contributing to the differences compared to our results. In contrast to these studies, we performed a precise, yet straightforward analysis and evaluation easily employable in clinical pathology.

Additionally, we analyzed (sentinel) lymph nodes with and without metastasis and addressed different tumor types, not having been focused on by van den Eynden et al. [13]. We did not observe a significant influence of the tumor type on metastasis-associated

lymphangiogenesis, with a comparable proportion especially of invasive lobular carcinoma in our and van den Eynden's study populations [12, 13, 31].

Possibly, focusing on patients with positive SLN helps in identifying lymph nodes with lymphangiogenesis by catching a favorable time point. In our cohort, only 30 nodal positive patients were included against 65 in van den Eynden's study [13]. Nevertheless, we meticulously analyzed a total of 63 tumor-draining lymph node specimens without micro- or macrometastasis and were not able to detect any lymphatic vessel formation. Lymphangiogenesis after neoadjuvant therapy may involve different biological processes rendering the results for neoadjuvant-treated colorectal patients difficult to transfer on primarily surgically resected BC samples, even for nodal negative patients [20].

Consistent with findings on littoral cells only briefly mentioned by van den Eynden et al. [12], we detected D2-40 staining of sinus endothelial/virgultar cells with faint to strong intensity which we assessed semiquantitatively as to intensity and proportion of stained sinus. Essentially in line with our previous study [18] investigating only few lymph node locations, we here found variable, partly strong D2-40 positivity in the subcapsular sinus, suggesting advanced lymphatic endothelial differentiation. Apparently, sinus endothelial cells may differentiate, to a variable extent, towards a more complete lymphatic endothelial phenotype as indicated by the expression of this specific marker of lymphatic vessel endothelium. Staining intensity and proportion of D2-40-stained subcapsular sinus correlated in both SLN and NSLN statistically significantly, signifying that strongly stained subcapsular sinus usually involved large parts of the total subcapsular sinus. A proportion of $\geq 35\%$ of subcapsular sinus stained for D2-40 in SLN correlated with and proved to be a prognostic parameter for worse OS, independent from tumor size, nodal status and age at surgery. Our data is limited with respect to the potential influence of receptor status and patient treatment, since no detailed information was available regarding possible adjuvant therapy. Also, it is yet too early for conclusions regarding the clinical significance of the concordance between staining intensity and proportion.

Recently, new technologies have fostered the discovery of a surprising heterogeneity of sinusoidal lymphatic endothelial cells with at least six different subpopulations and a subdivision into ceiling, floor and transversing cords for the subcapsular sinus [32–34]. However, histological D2-40/ podoplanin positivity has not been quoted by Jalkanen et al for cells lining human lymphatic sinus, but D2-40/ podoplanin positivity identifies lymphatic endothelial cells in lymph nodes by flow cytometry or is part of their gene expression profile [32]. In mouse subcapsular sinus, floor lymphatic endothelial cells are D2-40/ podoplanin positive [32]. Single reports are available for faint and focal staining of the subcapsular sinus for D2-40/ podoplanin in non-metastatic lymph nodes [12, 18], but this observation has not been further evaluated. Thus, our results fit into an increasingly complex picture of lymph node structure and its potential functional and prognostic implications, with its exact function yet to be disclosed. Care must be taken to differentiate diligently between features specific for mouse and human architecture. Further studies, including prospective validation, are necessary to unravel the biological background of D2-40/ podoplanin variations of the sinus endothelium and its full clinical relevance for human beings, particularly for breast cancer patients.

## Conclusions

Lymphangiogenesis was detected in association with BC metastases, with newly formed lymphatic vessels being significantly more numerous in larger metastases. However, intranodal lymphangiogenesis could not be verified in tumor-free regional lymph nodes of BC patients. SLN and NSLN partly demonstrated a strong and uniform positivity of sinus endothelial cells

of the subcapsular sinus for the lymphatic endothelial marker D2-40 with pronounced heterogeneity. A higher proportion of D2-40 positivity of the subcapsular sinus in SLN was associated with worse OS and emerged as independent prognostic parameter in multivariate analysis, fitting into the increasingly complex picture of lymph node structure and its functional implication for cancer spread and suggesting variable formation of a complete lymphatic endothelial phenotype with yet unknown biological significance. In contrast to other studies, we performed a precise, yet straightforward analysis and evaluation easily employable in the daily routine of clinical pathology. The concept of intranodal lymphangiogenesis remains to be further elucidated in the setting of clinical pathology including its potential prognostic impact.

## Supporting information

**S1 Table. Association of the type of lymph node metastasis with tumor type in SLN and NSLN.** LN: lymph node, ITC: isolated tumor cells, SLN: sentinel lymph node, ILC: invasive lobular carcinoma, NST: invasive carcinoma of no special type, tubular: tubular carcinoma, NSLN: non sentinel lymph node, NA: not applicable.
(DOCX)

**S2 Table. Association of CC category with tumor type in SLN and NSLN.** CC: Chalkley count, SLN: sentinel lymph node, ILC: invasive lobular carcinoma, NST: invasive carcinoma of no special type, NSLN: non sentinel lymph node.
(DOCX)

**S3 Table. Association of the categorized proportion of D2-40 stained sinus in SLN with clinicopathological parameters; categorical variables: Absolute (relative) frequencies, p values of the $\chi^2$- or Fisher's exact-tests for association with clinicopathological parameters; continuous variables: Mean (standard deviation), p values of the t test for association with clinicopathological parameters.** NST: invasive carcinoma of no special type, tubular: tubular carcinoma, LN: lymph node, ITC: isolated tumor cells.
(DOCX)

**S1 Fig. Flowchart for number of paraffin blocks with SLN evaluated at each stage.** ITC: isolated tumor cells, LVD: lymphatic vessel density.
(TIF)

**S2 Fig. Flowchart for number of paraffin blocks with NSLN evaluated at each stage.** LVD: lymphatic vessel density.
(TIF)

**S3 Fig. Distribution and survival based cutoff optimization for proportion of D2-40 stained sinus with respect to overall survival by "Cutoff Finder" algorithm.** (A) Histogram of proportion of D2-40 stained sinus in the 95 patients analyzed. The vertical line designates the optimal cutoff derived from the model. (B) For each possible cutoff, proportion of D2-40 stained sinus is correlated with overall survival. The hazard ration (HR) including 95% confidence interval (CI) is plotted in dependence of the cutoff. A vertical line designates the dichotomization showing the most significant correlation with survival. (C) The mean survival time is estimated in samples where proportion of D2-40 stained sinus is high and low, respectively. The difference of the mean survival times including 95% CI is plotted.
(TIF)

## Acknowledgments

We thank Angela Hartmann (passed away in July 2020) for expert technical assistance and Prof. Carsten Denkert for valuable discussion.

## Author Contributions

**Conceptualization:** Christina C. Westhoff, Roland Moll.

**Data curation:** Christina C. Westhoff, Matthias Kalder.

**Formal analysis:** Sabrina K. Müller, Paul Jank, Matthias Kalder, Roland Moll.

**Funding acquisition:** Christina C. Westhoff.

**Investigation:** Christina C. Westhoff, Sabrina K. Müller, Roland Moll.

**Methodology:** Roland Moll.

**Resources:** Roland Moll.

**Software:** Paul Jank.

**Supervision:** Christina C. Westhoff, Roland Moll.

**Validation:** Paul Jank.

**Visualization:** Christina C. Westhoff, Sabrina K. Müller, Paul Jank, Roland Moll.

**Writing – original draft:** Christina C. Westhoff.

**Writing – review & editing:** Sabrina K. Müller, Paul Jank, Matthias Kalder, Roland Moll.

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
