## [Decision Letter · Decision Letter 0]

18 May 2022

PONE-D-21-25472Nodal lymphangiogenesis and immunophenotypic variations of sinus endothelium in sentinel and non-sentinel lymph nodes of invasive breast carcinomaPLOS ONE

Dear Dr. Westhoff,

Thank you for submitting your manuscript to PLOS ONE. After careful consideration, we feel that it has merit but does not fully meet PLOS ONE’s publication criteria as it currently stands. Therefore, we invite you to submit a revised version of the manuscript that addresses the points raised during the review process.

Your manuscript has been assessed by two expert reviewers, whose comments are appended below. The reviewers have raised concerns regarding several aspects of the study methodology, as well as the framing of the objectives and discussion. Please ensure you address all of these points carefully in your revised manuscript.

We look forward to receiving your revised manuscript.

Kind regards,

Joseph Donlan

Editorial Office

PLOS ONE

Journal Requirements:

2.In your Data Availability statement, you have not specified where the minimal data set underlying the results described in your manuscript can be found. PLOS defines a study's minimal data set as the underlying data used to reach the conclusions drawn in the manuscript and any additional data required to replicate the reported study findings in their entirety. All PLOS journals require that the minimal data set be made fully available. For more information about our data policy, please see http://journals.plos.org/plosone/s/data-availability.

Reviewers' comments:

Reviewer's Responses to Questions

**Comments to the Author**

1. Is the manuscript technically sound, and do the data support the conclusions?

Reviewer #1: Yes

Reviewer #2: Partly

2. Has the statistical analysis been performed appropriately and rigorously? 

Reviewer #1: Yes

Reviewer #2: Yes

3. Have the authors made all data underlying the findings in their manuscript fully available?

Reviewer #1: Yes

Reviewer #2: Yes

4. Is the manuscript presented in an intelligible fashion and written in standard English?

Reviewer #1: Yes

Reviewer #2: Yes

5. Review Comments to the Author

Reviewer #1: Firstly, thank you for rewiew chance in your journal. I am lucky to review an article in my field of interest. It took me some time to scan the breast-related molecular pathology literature to be able to evaluate it. As for the article: I think the writing language of the article is of good quality, I find the relationship between the discussion and the results sufficient, the study is well described in the method section. I think the tables sufficiently support the text integrity.

Reviewer #2: Westhoff et al. have done a clinicopathological study in 95 patients on lymphovascular changes that are associated with BC nodal metastases using the D2-40 marker. They observed lymphangiogenesis only in lymph nodes with metastases and the number of vessels correlated positively with the size of metastases (p < 0.001) using 2 morphometric methods with good concordance. This contradicts previous studies which seemed to suggest that lymphangiogenesis precedes lymph nodal metastasis of BC. However, they observed D2-40 staining in the subcapsular sinus, where staining intensity and stained proportion correlated both in SLN and NSLN (p < 0.001). A higher proportion of D2-40 stained subcapsular sinus in SLN was significantly associated with worse overall survival (p= 0.0036) and an independent prognostic parameter in multivariate analysis (p= 0.03, HR 2.89).

My comments:

1. The authors need to state the objective of the current study clearly. In their own words

“We now aimed to evaluate these findings in a typical patient cohort of a BC center employing the sentinel node biopsy since 2000. An important objective was the feasibility with routine immunohistochemical repertoire,”

“Differentiation for tumor types, especially for invasive lobular carcinoma, was also important to us”

“A special emphasis was put on the analysis of the subcapsular sinus endothelium and the possible clinical relevance of its podoplanin expression.”

Though they have expressed a general direction of inquiry, the exact objectives of the study need to be laid out more clearly in unambiguous statements.

2. Methodology

a.Why were archived specimens from 16-20 years ago chosen for analysis and not recent ones?

b.The authors need to mention in the statistical methodology section, what was the null hypothesis. What was the expected percentage staining in NSLN nodes, and the expected difference between SLN and NSLN in terms of lymphangiogenesis within metastases and in the subcapsular sinus?

c.What was the rationale for choosing a multivariate model with only the following as independent variables? age at surgery, pN stage, tumor size. Why were other prognostic variables not included (eg. T stage, ER, PR, HER2 status, grade)

d.The authors have not taken into account the treatment patient received when assessing factors influencing survival. The receptor status and treatment received in terms of antiHER2 directed therapy, anthracycline and taxane are very important prognostic factors for survival which cannot be ignored. They should either mention these as a major drawback of the survival analysis or give data regarding the above.

e.Wherever continuous variables are compared using Pearson or Spearman, the correlation coefficient should be mentioned alongside the p-value in the text for the readers to understand the strength of association.

3.“In contrast to other studies, we performed a precise, yet straightforward analysis and evaluation easily employable in the daily routine of clinical pathology”.

The authors need to elucidate the clinical importance of demonstrating lymphangiogenesis within LN with metastases. The KM curves for OS and the multivariate analysis are encouraging for podoplanin as a biomarker which may predict survival. This requires prospective validation, however.

4.The authors need to clarify the flowchart (STROBE) of the number of specimens studied with respect to SLN, NSLN and all cases. The following statements mention N but it is not clear to the reader. Perhaps the flow of N cases starting from archived specimens can be mentioned in the results or methodology, if not graphically represented.

-“Representative paraffin blocks of 75 SLN and/or non-SLN (NSLN) from n= 95 patients were recruited”

-“All cases of SLN and NSLN without micro- or macrometastasis in the present study, comprising 63 cases”

-“Among 95 SLN overall 134 investigated, 32 harbored metastases”

-“In 40 cases, additional NSLN could be investigated”

-“Established lymph node metastases in 20 SLN and five NSLN were available for further immunohistochemical analysis of newly formed D2-40 positive lymphatic vessels”

5. What was the association between the proportion of D2-40 stained subcapsular sinus in NSLN and overall survival?

6. The authors need to elucidate why the concordance between staining intensity and proportion of stained sinus is clinically important.

7. The authors could provide some data on the difference in D2-40 staining between breast cancer subtypes(ER, PR, HER2) if available. In the supplementary table, subcapsular sinus staining across subtypes is presented, however the association between subtype and lymphangiogenesis in metastasis is not mentioned.

P2 Line 22

Suggestion- change to “(podoplanin) “expression on” lymphatic vessels and the subcapsular sinus.”

P2 Line 24

Suggestion- Data was explored “with respect to”

Table 1

When mean is used as a measure of central tendency, the standard deviation is mentioned instead of range. This is applicable if the variable is distributed normally. This is unlikely to be the case with such small numbers, hence the better statistic to represent central tendency is median and IQR in most of the variables unless normal distribution is confirmed with a histogram or other methods.

In this table it would be better to mention the number of patients who were triple-negative(ER, PR and HER2neu negative) separately. This number cannot be obtained from the table.

Figure 6

Please provide population at risk table along with KM curve if possible

6. PLOS authors have the option to publish the peer review history of their article (what does this mean?). If published, this will include your full peer review and any attached files.

Reviewer #1: **Yes: **ONUR DÜLGEROĞLU

Reviewer #2: No

---

## [Author Response · Author response to Decision Letter 0]

3 Aug 2022

We thank the reviewers for their thorough revision and helpful remarks regarding our manuscript. We have responded to all questions within the document "response to reviewer" in detail and changed our manuscript accordingly.

---

## [Decision Letter · Decision Letter 1]

12 Jan 2023

Nodal lymphangiogenesis and immunophenotypic variations of sinus endothelium in sentinel and non-sentinel lymph nodes of invasive breast carcinoma

PONE-D-21-25472R1

Dear Dr. Westhoff,

We’re pleased to inform you that your manuscript has been judged scientifically suitable for publication and will be formally accepted for publication once it meets all outstanding technical requirements.

Kind regards,

Josh Thomas Georgy, MD

Guest Editor

PLOS ONE

Additional Editor Comments (optional):

Dear Christina C. Westhoff,

Thank you for submitting your manuscript, "Nodal lymphangiogenesis and immunophenotypic variations of sinus endothelium in sentinel and non-sentinel lymph nodes of invasive breast carcinoma," for consideration at PLOS ONE. We have reviewed your paper and are pleased to inform you that it has been accepted for publication.

Your study provides valuable insights into the presence of lymphangiogenesis in breast cancer-associated lymph nodes and its potential correlation with patient outcomes. Your findings that larger metastases have more lymphatic vessels and that staining intensity and proportion of D2-40 in the subcapsular sinus of SLN are associated with worse overall survival as an independent predictor are novel and clinically relevant. These findings will need to be verified in larger cohorts prospectively. I hope the publication of your paper will trigger the initiation of such studies in the future.

We appreciate your contributions to the field and look forward to publishing your paper in our journal.

Reviewers' comments:

Reviewer's Responses to Questions

**Comments to the Author**

1. If the authors have adequately addressed your comments raised in a previous round of review and you feel that this manuscript is now acceptable for publication, you may indicate that here to bypass the “Comments to the Author” section, enter your conflict of interest statement in the “Confidential to Editor” section, and submit your "Accept" recommendation.

Reviewer #3: All comments have been addressed

2. Is the manuscript technically sound, and do the data support the conclusions?

Reviewer #3: Yes

3. Has the statistical analysis been performed appropriately and rigorously? 

Reviewer #3: Yes

4. Have the authors made all data underlying the findings in their manuscript fully available?

Reviewer #3: Yes

5. Is the manuscript presented in an intelligible fashion and written in standard English?

Reviewer #3: Yes

6. Review Comments to the Author

Reviewer #3: The study is very interesting and done in a very meticulous manner. Findings are correlating with the data provided.

7. PLOS authors have the option to publish the peer review history of their article (what does this mean?). If published, this will include your full peer review and any attached files.

Reviewer #3: **Yes: **Dr. Elanthenral Sigamani

---

## [Editor Report · Acceptance letter]

16 Jan 2023

PONE-D-21-25472R1 

Nodal lymphangiogenesis and immunophenotypic variations of sinus endothelium in sentinel and non-sentinel lymph nodes of invasive breast carcinoma 

Dear Dr. Westhoff:

I'm pleased to inform you that your manuscript has been deemed suitable for publication in PLOS ONE. Congratulations! Your manuscript is now with our production department. 

Kind regards, 

on behalf of

Dr. Josh Thomas Georgy 

Guest Editor

PLOS ONE